# Effectiveness of sofosbuvir based direct-acting antiviral regimens for chronic hepatitis C virus genotype 6 patients: Real-world experience in Vietnam

**Dung Thanh Nguyen[1], Thanh Thi Thanh Tran[2], Ngoc My Nghiem[1], Phuong Thanh Le[1], Quang Minh Vo[1], Jeremy Day[2,3], Motiur Rahman**👤[2,3]*, **Hung Mạnh Le[1]**

**1** The Hospital for Tropical Diseases, Ho Chi Minh City, Vietnam, **2** Oxford University Clinical Research Unit, Wellcome Trust Asia Programme, The Hospital for Tropical Diseases, Ho Chi Minh City, Vietnam, **3** Centre for Tropical Medicine and Global Health, Nuffield Department of Medicine, Oxford University, Oxford, United Kingdom

* mrahman@oucru.org

**Data Availability Statement:** The data presented in the manuscript are extracted from the Hospital for Tropical Diseases, Ho Chi Minh City Vietnam and

## Abstract

### Background

Hepatitis C virus (HCV) genotype 6 is the commonest cause of chronic hepatitis C infection in much of southeast Asia, but data on the effectiveness of direct-acting antiviral agents (DAAs) against this genotype are limited. We conducted a retrospective cohort study of patients attending the Hospital for Tropical Diseases (HTD), Ho Chi Minh City, Vietnam, to define the effectiveness of DAAs in the treatment of chronic HCV genotype 6 in actual practice.

### Methods

We included all patients with genotype 6 infections attending our hospital between March 2016 and October 2017 who received treatment with sofosbuvir-based DAA treatment regimens, and compared their responses with those with genotype 1 infections.

### Results

1758 patients (1148 genotype 6, 65.4%; 610 genotype 1, 34.6%) were analyzed. The majority of patients (1480, 84.2%) received sofosbuvir/ledipasvir (SOF/LDV) ± ribavirin (RBV); 278 (15.8%) received sofosbuvir/Daclatasvir (SOF/DCV) ± RBV. The median age of the patients was 57 years, (interquartile range (IQR) 46–64 years) The baseline HCV viral load (log IU/ml) was significantly higher in patients infected with genotype 6 compared with those infected with genotype 1 (6.8, 5.3–6.6 versus 6.3, 5.3–6.5 log10 IU/ml, $p$ = <0.001, Mann Whitney U test). A sustained virological response (SVR), defined as an undetectable viral load measured between 12 and 24 weeks after completing treatment, and indicating cure, was seen in 97.3% (1711/1758) of patients. Treatment failure, defined as HCV viral load $\geq$15 IU/ml $\geq$12 weeks after completing treatment appeared to be more frequent in patients

property of Hospital for Tropical Diseases, Ho Chi Minh City Vietnam. Institutional Review Board (IRB) of the Hospital for Tropical Diseases approved the access to fully anonymized dataset for analysis to the investigators and Oxford University Clinical research Unit (OUCRU). All data presented in the manuscript can be accessed through "OUCRU data sharing policy" and request for access to data can be sent to DAC@oucru.org.

**Funding:** The author(s) received no specific funding for this work.

**Competing interests:** The authors have declared that no competing interests exist.

infected with genotype 6 virus (3.2%, 37/1148) than in those infected with genotype 1 (1.7%, 10/610), $p$ = 0.050 chi-squared test). We found no evidence that patient's age, gender, liver cirrhosis, diabetes, HBV or HIV coinfection, prior treatment failure with pegylated interferon therapy, body mass index (BMI), aspartate aminotransferase to platelet ratio index (*APRI)*, or fibrosis 4 (FIB-4) index were associated with treatment failure.

## Conclusions

Our study suggests that patients with HCV genotype 6 infection in Vietnam may respond less well to treatment with sofosbuvir based DAAs than patients with genotype 1 infections. Further studies are needed to confirm this observation and to define whether it is driven by genotype-specific mutations.

## Introduction

The Global Health Sector Strategy (GHSS) for viral hepatitis 2016–2021 calls for the elimination of viral hepatitis as a public health threat, reducing new infections by 90% and mortality by 65% by 2030 [1]. The WHO Western Pacific region including Vietnam bears the highest burden of Hepatitis C virus (HCV) globally, with approximately 19.2 million chronic infections [1]. The introduction of direct-acting antiviral agents (DAAs) has revolutionized HCV treatment and increasing numbers of patients are being treated. Several phase III clinical trials (Neutrino, Fission, and Valence) have demonstrated the efficacy, simplicity, and tolerability of DAAs [2–4] in the treatment of HCV in well-resourced setting. The sustained virologic response rate (SVR), defined as an undetectable viral load 12 weeks after completion of treatment and considered to represent cure, is consistently above 90% for most HCV-infected patient populations [5, 6].

HCV is classified into 7 genotypes, and these have specific geographical distributions. HCV genotype 6 is largely confined to China and southeast Asia, including Vietnam, Laos, Cambodia, Myanmar, Taiwan and the southern Chinese provinces of Guangxi, Guangdong and Hainan [7,8]. Thirty-one subtypes of genotype 6 have been recognized in the region, indicating a local emergence and evolution [9]. In the south of Vietnam, up to 60% of HCV infections are caused by genotype 6 [10] and therefore the success of HCV elimination in the region depends upon the effectiveness of DAA combinations against this genotype. While a number of DAAs, such as Sofosbuvir (SOF), are believed to have antiviral effects that are independent of the virus genotype, there are limited data on the efficacy of DAA treatments for HCV genotype 6 infections. This reflects the limited numbers of patients with the genotype recruited into clinical trials [11, 12]. While studies from New Zealand and Hong Kong that have included small numbers of genotype 6 infected patients suggest that SOF-based regimens, including SOF + Ledipasvir (LDV) and SOF+ Ribavirin (RBV), are likely to be effective for most cases [13, 14], few data exist regarding the efficacy of treatment in resource poor settings.

The 'real world' effectiveness of medical treatments do not necessarily reflect their efficacy rates seen in clinical trials, and HCV infection is no exception [15, 16]. These differences likely reflect heterogeneities in patient characteristics, clinical practice, resources, care coordination, treatment drug combinations, and treatment adherence and duration, and become apparent only when a drug is prescribed to a wider population [5]. Real-world data on the effectiveness of DAAs in HCV genotype 6 infections from the geographic locations where it is prevalent are

particularly limited [17]. Understanding the effectiveness of DAAs in such settings and in normal practice is crucial to inform policymakers when designing HCV treatment programs.

Vietnam is among the top 20 countries with the highest HCV burdens, with a population seroprevalence of between 0.9% and 1.2% [18]. DAAs have been the recommended first line treatment for HCV infection in Vietnam since 2016 [19]. All provincial hospitals and selected referral HCV treatment centers, including the Hospital for Tropical Diseases (HTD), Ho Chi Minh City, have prescribed DAA treatment since then, although the cost of this has been met by patients [17]. We present here our experience of the use of two sofosbuvir-based DAA regimens (SOF/LDV ± RBV, and SOF/Daclatasvir (DCV) ± RBV) in treatment-naïve patients infected with HCV genotype 1 or 6.

## Materials and methods

### Study description and ethical approval

We performed a retrospective, intent-to-treat cohort analysis of all chronic HCV (genotype 1 and 6) infected patients attending our hospital who began treatment with DAA combination therapy between March 2016 and October 2017. We include only DAA-inexperienced patients in the study; however, we did include patients who had previously failed to respond to treatment with non-DAA treatment history (i.e. Pegylated interferon (PegINF) and RBV). All DAA treatment was prescribed through the hospital pharmacy. To be included patients had to be age ≥18 years, infected with HCV genotype 1 or 6, and have initiated treatment with either SOF/LDV ± RBV, or SOF/DCV ± RBV. We excluded patients where the baseline and/or SVR HCV viral load data were unavailable including patients with incomplete treatment. The study received ethical approval from the Ethics Review Committee of the Hospital for Tropical Diseases (approval no CS/ND/16/02 date 23/11/2017).

### Setting, patients and data extraction

The Hospital for Tropical Diseases (HTD), Ho Chi Minh City, is a 650-bed infectious disease hospital, and a designated specialized care provider and referral centre for patients with infectious hepatitis from the centre and south of Vietnam [10]. In 2015 HTD introduced an electronic record keeping system for the outpatients departments. These records include sociodemographic, clinical, imaging, prescribing, diagnostic and treatment outcome data for each patient under a unique identification number (ID). The HTD clinical laboratory maintains a separate database of all laboratory investigations conducted on patient samples; laboratory data are stored using a separate laboratory number linked to the unique patient ID. For this study, the hospital database was screened for the diagnosis of chronic HCV infection and treatment with DAAs. The hospital records management team extracted sociodemographic, clinical, laboratory and drug prescription and treatment outcome data for all eligible patients from the database according to the study proforma. All patient data were anonymized by replacing the patient identifier (unique ID) with a unique study number before transfer to the study investigators.

### Data variables

Baseline variables were defined as the most recently available data prior to the initiation of DAA treatment (IOT). These included age, sex, geolocation (to the district level), liver cirrhosis status, diabetes, HIV and HBV infection status, hemoglobin, white blood count (WBC), platelet count, bilirubin, albumin, gamma-glutamyltransferase (GGT), alanine aminotransferase (ALT), aspartate aminotransferase (AST), alpha-fetoprotein (AFP), blood glucose, HCV

viral load and genotype, and Fibroscan score. Blood counts and biochemical tests were ascertained using a Sysmex XN-100 analyzer (Sysmex USA) and a Cobas 6000 analyzer (Roche, Basel, Switzerland) in the HTD clinical laboratory (ISO 15189; 2012 certified). Liver fibrosis was estimated using an Abbott FibroScan VCTE, (Abbott, Chicago, IL, USA). HCV viral load was measured using a commercial real-time polymerase chain reaction assay (COBAS AmpliPrep COBAS Taqman HCV Test version 2.0; Roche Molecular Diagnostics, Pleasanton, CA, USA), which defines a HCV viral load ≤15 IU/mL as "undetectable". HCV genotype was determined by "Real-time HCV Genotype assay II" using an Abbott m2000sp/rt system (Abbott Molecular Inc, Chicago, IL, USA). The fibrosis 4 (FIB-4) values were calculated using the formula age (years) $\times$ AST [U/l]/(platelets [$10^9$/l] $\times$ (ALT [U/l])$^{1/2}$). The APRI values were calculated using the formula "AST/upper limit of normal]/platelet count [$10^9$/L]" $\times$ 100 (the upper limit of normal of AST in our hospital is 37 IU/L for women and 40 IU/L for men) [20]. A FIB-4 <1.45 indicates absence of fibrosis and >3.25 indicates cirrhosis; an APRI score <0.5 indicates absence of fibrosis, >1.5 indicates fibrosis and >2.0 indicates cirrhosis [21].

## Treatment

Patients received treatment according to the Vietnamese national guidelines at the time, summarised in S1 Table [22]. In line with these guidelines, the treatment regimen was selected based upon the HCV genotype and the presence or absence of cirrhosis. For patients without cirrhosis, the guidelines recommended a treatment duration of 12 weeks. Patients with cirrhosis were treated with either SOF/LDV or SOF/DCV for 24 weeks, or either of these combinations together with RBV for 12 weeks. The choice regarding these latter regimens was made by the physician in discussion with the patient.

SOF/LDV was given daily as a single oral fixed dose combination tablet consisting of 400 mg SOF and 90 mg LVD (sourced from any of Mylan Laboratories, India, Hetero Laboratories, India and M/s Natco Pharma, India). SOF/DCV was given as 400mg SOF and 30mg DCV once daily (sourced from Mylan Laboratories, India, Hetero Laboratories, India and M/s Natco Pharma, India). The treatment duration received by each patient was confirmed by review of prescriptions and the number of cumulative days' supply purchased by each patient, with purchase of medication for either 84 (12 weeks) or 168 days (24 weeks) from the hospital pharmacy being assumed to indicate the completion of 12 or 24 weeks of treatment respectively. We calculated the end of treatment (EOT) as the last day covered by the prescription related to the initial date of medication dispensing by the hospital pharmacy, cross-checked with the number of tablets bought. At HTD all patients receive advice on the importance of treatment adherence as per standard of care at each visit. Where doses are missed they are recommended to take the missed dose if within 16 hours of the due time. If more than 16 hours have elapsed, they are recommended to take the next dose at the due time.

## Treatment outcome monitoring

HCV viral load was measured before IOT, at weeks 4, week 8 (if the HCV viral load was detectable at week 4), and either 12 or 24 weeks after the end of treatment (EOT, see S2 Table) [22]. Rapid virologic response (RVR) was defined as an HCV RNA <15 IU/mL 4 weeks after IOT. Treatment success—sustained virological response (SVR) was defined as unquantifiable HCV RNA (LOD <15 IU/mL) on all HCV RNA tests measured from 12 weeks or 24 weeks after the EOT or undetectable HCV RNA on last HCV RNA test 12 weeks or 24 weeks after EOT. Failure to achieve an SVR at 12 or 24 weeks after the EOT was defined as treatment failure. We defined breakthrough and relapse of infection as the achievement of an undetectable HCV

RNA during treatment, followed by the detection of HCV RNA $\geq$15 IU/mL while on treatment (breakthrough), or after treatment completion (relapse).

## Data analysis

Data analysis was performed using Statistical Package for Social Science (SPSS) software (IBM SPSS Statistics 23, NY USA). The main outcome of interest was the response to treatment. We analysed the success of the treatment on an intent to treat basis (n = 1758). Baseline descriptive statistics were summarized for the variables of interest. Comparisons between groups were performed using either the chi-squared or Fisher's exact tests for categorical variables; *t*-tests and the Mann-Whitney U-test were used for continuous variables. We used logistic regression to determine the baseline factors associated with SVR. A two-sided *P* value of $\leq$0.05 was considered statistically significant.

## Results

### Enrollment

From March 2016 to October 2017, 2817 patients infected with HCV attended the outpatient department at the HTD and initiated treatment with DAAs. Among these 369 patients had either genotype 2 or 3 infections and therefore were excluded from the analysis. Of the remaining 2448 patients, 684 were excluded because either baseline or SVR HCV viral load data were missing. Additionally, six patients treated with Elbasvir and Grazoprevir were excluded, resulting in 1758 patients available for analysis (Fig 1). Among the 1758 patients, 34.6% (610/1758) had genotype 1 infections and 65.4% (1148/1758) had genotype 6 infections. The 1758 patients had 4959 outpatient visits after IOT; 1401 patients (79.7%) had at least three more visits. HCV viral load at 4 weeks after IOT was available for 98.9% (1739/1758) of patients. Viral load at 12 week post-EOT only, both at 12 and 24 weeks post-EOT and at 24 weeks post-EOT only were available for 46.3% (814/1758), 44.8% (788/1753) and 8.9% (156/1758) patients respectively.

### Demographics

Table 1 presents the baseline demographic and biochemical characteristics of the study population categorized by HCV genotype. The age of the patients (median; interquartile range (IQR)) was 57.0; 46–64 years. Patients with genotype 6 infections were slightly older than those with genotype 1 infections (mean age) 55.87 years versus 53.20 years; *p* = <0.001, Mann Whitney U test). Overall there was a preponderance of female patients 56.9% (1001/1758). There was a preponderance of women amongst the genotype 6 infected cohort where they accounted for 59.8% of patients (95% confidence interval (CI) 57.0–62.6%; men 40.2%, 95%CI 37.4–43.0%, N = 1148). There was no difference in gender distribution amongst genotype 1 infections (women 51.5%, 95%CI 47.5–55.4%; men 48.5%, 95%CI 44.6–52.5%, N = 610). There was evidence of cirrhosis in 35.4% (622/1758) of patients and there was no difference in prevalence of liver cirrhosis between genotype 1 and 6 infected patients (*p* = .064, chi-squared test). There was a higher prevalence of HIV infection amongst patients with HCV genotype 1 infection than amongst patients with HCV genotype 6 infection (2.5% (15/610) versus 0.9% (10/1148) *p* = 0.008, chi-squared test) patients. There was no signification difference in HBV coinfection among genotype 1 and 6 patients (2.8%; (17/610) versus 2.7% (31/1148) *p* = 0.0531, chi-squared test). We found that markers of liver inflammation AST, ALT, AFP, GGT were statistically significantly higher in patients with genotype 1 infection, although the actual differences were small. In contrast, the HCV viral load was significantly higher in patients infected with genotype 6 virus compared with genotype 1 virus (6.6± 6.8 versus 6.3

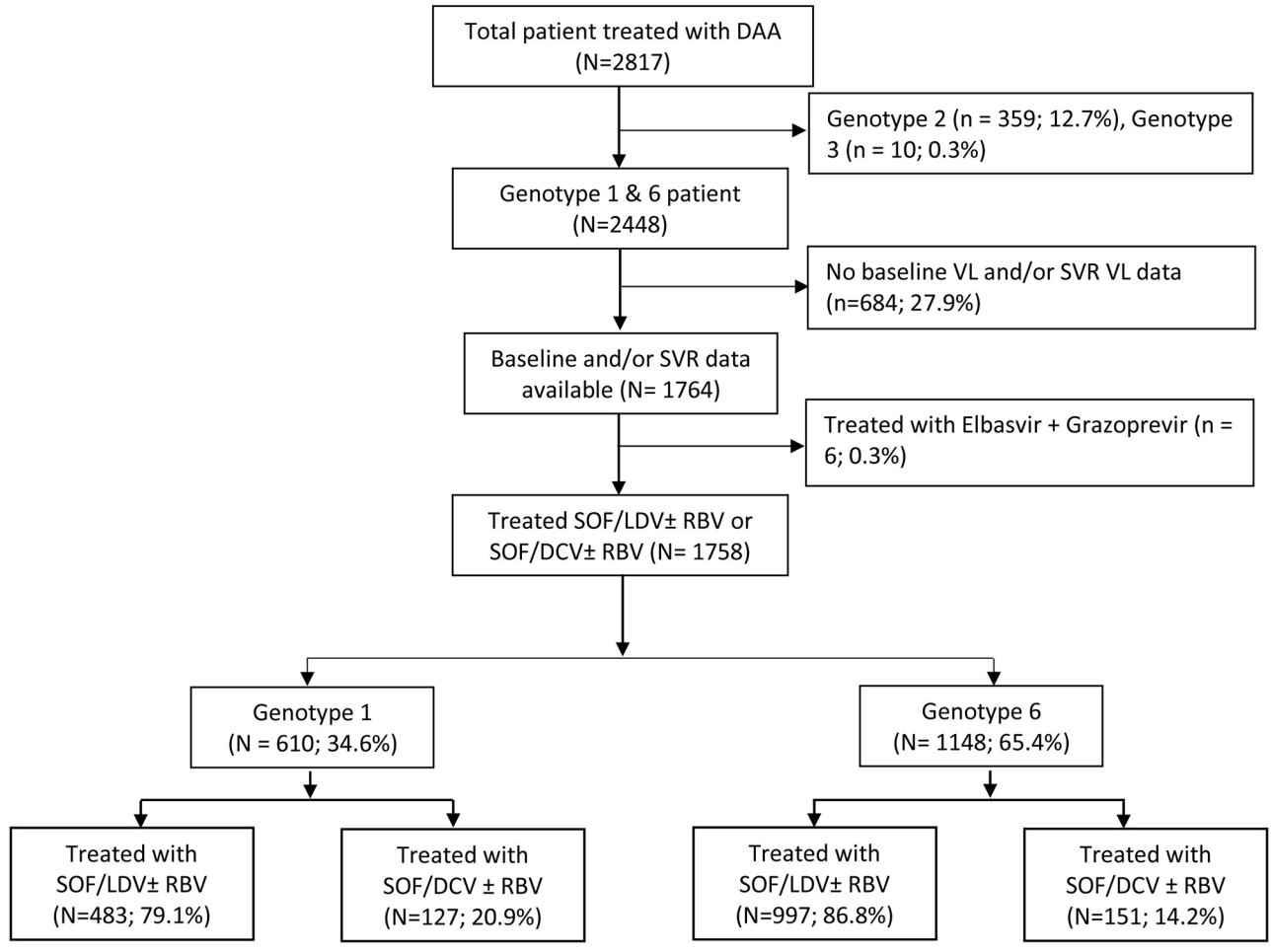

**Fig 1. Enrollment and analysis of patients.**

±6.5, $p$ = <0.001, Mann Whitney U test)). There was no significant difference in APRI and FIB-4 scores between patients infected with genotype 1 versus genotype 6. Patients infected with genotype 1 were more likely to have had a prior treatment failure episode with PegINF/ RBV (7.0% (43/610) versus 4.7% (54/1148); $p$ = 0.04, chi-squared test).

## Treatment

Details of the treatment prescribed are shown in Table 1. Most patients (84.2%, 1480/1758) were treated with SOF/LDV± RBV; 15.8% (278/1758) were treated with SOF/DCV ± RBV. Patients with genotype 1 infections were more frequently treated with SOF/DAC±RBV than patients with genotype 6 infections (20.8% (127/610) versus 13.2% (151/1148); $p$ = <0.001, chi-squared test). The majority of patients with cirrhosis received treatment with ribavirin and DAAs for 12 weeks (compensated cirrhosis: 90.0% (522/613); decompensated cirrhosis 66.6% (6/9)). There was no difference in the use of the ribavirin-sparing 24 week treatment regimen between genotypes 1 and 6 patients (genotype 1: 4.8% (29/610) versus genotype 6: 3.0% (35/ 1148)). Patients who received 24 week treatment were significantly older (median; ±IQR) (60; 55–65 years versus 56; 46–64 years, $p$ = 0.003, Mann Whitney U test), had higher liver stiffness

**Table 1. Baseline characteristics of all patients, and genotype 1 and 6 patients.**

| | All patient | Genotype 1 | Genotype 6 | *P* value |
|---|---|---|---|---|
| | N = 1758 | N = 610 (34.7%) | N = 1148 (65.3%) | |
| Age (years) [β] | 57; 46–64 | 55; 43–62 | 57; 48–65 | < .001[$, ***] |
| <40 [δ] | 15.6 (274) | 20.5 (125) | 13.0 (149) | |
| 41–55 [δ] | 30.9 (544) | 30.5 (186) | 31.2 (358) | |
| >55 [δ] | 53.5 (940) | 49.0 (299) | 55.8 (641) | |
| Gender [δ] | | | | < .001[Ω, ***] |
| Female | 56.9 (1001) | 51.5 (314) | 59.8 (687) | |
| Male | 43.1 (757) | 48.5 (296) | 40.2 (461) | |
| Liver [δ] | | | | 0.064[Ω] |
| Non-cirrhosis | 64.6 (1136) | 62.5 (381) | 65.8 (755) | |
| Compensated Cirrhosis | 34.9 (613) | 36.6 (223) | 34.0 (390) | |
| Decompensated Cirrhosis | 0.5 (9) | 1.0 (6) | 0.3 (3) | |
| Diabetes [δ] | 3.1 (55) | 4.1 (25) | 2.6 (30) | 0.610[Ω] |
| HBV coinfection [δ] | 2.7 (48) | 2.8 (17) | 2.7 (31) | 0.531[Ω] |
| HIV coinfection [δ] | 1.4 (25) | 2.5 (15) | 0.9 (10) | 0.008[Ω, **] |
| BMI (kg/m2) [α] | 22.73 ± 3.27 (13.42–38.89) | 22.79 ±3.25 | 22.64 ±3.28 | 0.454[$] |
| <18 [δ] | 5.0 (70) | 4.3 (21) | 5.3 (49) | |
| 18–25 [δ] | 73.7 (1033) | 73.3 (355) | 73.9 (678) | |
| >25 [δ] | 21.3 (299) | 22.3 (108) | 20.8 (191) | |
| Fibroscan (Kpa) [α] | 12.77 ± 11.06 | 13.22±11.99 | 12.52±10.53 | 0.732[$] |
| ALT (U/L) [α] | 71.38 ± 58.52 | 75.8 ± 59.8 | 69.0 ± 57.7 | 0.003[$, **] |
| AST (U/L) [α] | 62.76 ± 43.74 | 64.9 ± 41.3 | 61.5 ± 44.9 | 0.015[$] |
| Bilirubin (µmol/L) [α] | 7.70 ± 17.84 | 10.1 ± 27.0 | 6.3 ± 9.1 | 0.739[$] |
| Creatinine (µmol/L) [α] | 72.74 ± 15.45 | 73.9 ± 15.4 | 72.0 ± 15.4 | 0.030[$, *] |
| Albumin (g/L) [α] | 40.63 ± 4.15 | 40.5 ± 4.2 | 40.6 ± 4.0 | 0.863[$] |
| AFP (ng/ml) [α] | 14.90 ± 42.76 | 23.1 ± 64.9 | 10.7 ± 23.9 | 0.001[$, **] |
| GGT (U/L) [α] | 71.41 ± 80.94 | 82.8 ± 98.3 | 65.3 ± 69.2 | 0.001[$, **] |
| Glucose (mmol/L) [α] | 5.88 ± 1.74 | 5.9 ± 2.0 | 5.8 ± 1.5 | 0.232[$] |
| HCV RNA [δ] (log IU/ml) | 6.5; 5.3–6.5 | 6.3, 5.3–6.5 | 6.8, 5.3–6.6 | 0.001[$, **] |
| ≤6X10$^6$ IU/mL [δ] | 45.5 (800) | 48.9 (298) | 43.7 (502) | |
| >6X10$^6$ IU/mL [δ] | 54.5 (958) | 51.1 (312) | 56.3 (646) | |
| APRI score [α] | 6.8 ± 23.8 | 6.48 ± 23.1 | 7.05 ± 24.0 | 0.319[$] |
| <2 [δ] | 5.0 (83) | 5.4 (31) | 4.8 (52) | |
| >2 [δ] | 95.0 (1565) | 94.6 (543) | 95.2 (1022) | |
| FIB-4 score [α] | 8.9 ± 23.4 | 8.54 ± 22.8 | 9.21 ± 23.8 | 0.350[$] |
| <3.5 [δ] | 25.1 (427) | 24.4 (146) | 26.2 (281) | |
| >3.5 [δ] | 74.1 (1221) | 70.2 (428) | 73.8 (793) | |
| Prior Therapy failure [δ] | 5.5 (97) | 7.0 (43) | 4.7 (54) | 0.040[Ω, *] |
| Treatment regimen [δ] | | | | |
| SOF/LDV ± RBV | 84.2 (1480) | 79.2 (483) | 86.8 (997) | <0.001[Ω, ***] |
| SOF/DAC ± RBV | 15.8 (278) | 20.8 (127) | 13.2 (151) | |
| Treatment duration (all patient) [δ] | | | | |
| 12 week | 94.6 (1694) | 95.2 (581) | 97.0 (1113) | 0.069[Ω] |
| 24 week | 3.6 (64) | 4.8 (29) | 3.0 (35) | |
| Treatment duration (compensated cirrhosis) [δ] | | | | |
| 12 week | 90 (552/613) | 88.3 (197/223) | 91.0 (355/390) | 0.326[Ω] |
| 24 week | 10 (61/613) | 11.7 (26/223) | 9.0 (35/390) | |

(*Continued*)

**Table 1.** (Continued)

| | All patient | Genotype 1 | Genotype 6 | *P* value |
|---|---|---|---|---|
| | N = 1758 | N = 610 (34.7%) | N = 1148 (65.3%) | |
| Treatment duration (decompensated cirrhosis) [δ] | | | | |
| 12 week | 66.6 (6/9) | 50 (3/6) | 100 (3/3) | 0.464 [Ω] |
| 24 week | 33.3 (3/9) | 50 (3/6) | 0.0 (0/3) | |
| Received Ribavirin [δ] | | | | |
| Yes | 34.6 (608) | 35.4 (216) | 34.1 (392) | 0.596 [Ω] |
| No | 65.4 (1150) | 64.6 (394) | 65.9 (756) | |
| Rapid Virological Response achieved [δ] | | | | 0.728 [Ω] |
| Yes | 88.2 (1533) | 87.4 (528) | 88.5 (1005) | |
| No | 11.8 (206) | 12.6 (76) | 11.5 (130) | |

[α]: mean ±SD;

[β]: median; interquartile range;

[δ]: %(n)

[Ω]: Chi—square test;

[$]: Mann Whitney U test;

[*]: $p = 0.01–0.05$;

[**]: $p \leq 0.001–0.05$;

[***]: $p < 0.001$.

(Kpa 30.9 ± 17.5 versus 12.2 ±10.3; $p = <0.001$, Mann Whitney U test), higher APRI score (6.8 ± 23.9 versus 6.2 ± 20.8; $p = <0.001$, Mann Whitney U test), and higher FIB4 score (11.7 ± 20.9 versus 8.8 ± 23.5; $p = <0.001$, Mann Whitney U test) compared to patients who received 12 week treatment.

## Effectiveness

Overall, 88.2% (1533/1739) of patients had RVR, (undetectable HCV viral loads <15 IU/ml) by 4 weeks after IOT (Genotype 1: 87.4%, 528/604; genotype 6: 88.5%, 1005/1135, $p = 0.54$ chi-squared test, Table 1). There was no significant difference in RVR rates between patients treated with SOF/LDV ± RBV versus SOF/DCV ± RBV (88.4%, 1293/1463 versus 87.0% 240/276, $p = 0.280$ chi-squared test). A RVR was significantly more likely to be seen in patients with baseline HCV viral loads <6, 000,000 IU/ml than in those with viral loads ≥ 6, 000, 000 IU/mL (95.2%; 752/790 versus 82.3%; 781/949, $p = < 0.001$ chi-squared test).

Final treatment outcomes are shown in Table 2. SVR was achieved in 97.3% patients (1711/1758). Treatment failure was observed in 2.7% (47/1758) patients overall, and may have been slightly more common in patients with genotype 6 infections (3.2%; 37/1148 in genotype 6 infections versus 1.6%; 10/610 in genotype 1, $p = 0.050$ chi-squared test) Table 2. We could not detect any influence of age, gender, cirrhosis, diabetes, HBV or HIV coinfection, BMI, APRI or FIB-4 index on the likelihood of achieving SVR. Patients who experienced treatment failure tended to have higher baseline viral loads than patients who experienced treatment success (6.7 ± 7.0 versus 6.4 ± 6.7 log IU/ml) although this is not statistically significant ($p = 0.055$, Mann Whitney U test). 3.3% (32/958) of patients with baseline viral load >6,000,000 IU/ml had eventual treatment failure compared to 1.9% (15/800) with <6,000,000 IU/ml ($p = 0.058$, chi-squared test). Attainment of a RVR did not predict an increased likelihood of eventual treatment success (SVR achieved in 97.4% (1493/1533) of patients who had RVR versus in 97.6%; (201/206) of patients who did not, $p = 0.877$, chi-squared test (Table 2).

**Table 2. Comparison of baseline characteristics in patients according to eventual treatment outcome (sustained virological response, SVR for 1758 patients receiving SOF/LED ± RBV or SOF/DAC ± RBV.**

| Variable | SVR achieved | SVR failure | P value |
|---|---|---|---|
| | n = 1711 | n = 47 | |
| | % (n) | % (n) | |
| Genotype [δ] | | | **0.050** |
| Genotype 1 | 98.4 (600/610) | 1.6 (10/610) | |
| Genotype 6 | 96.8 (1111/1148) | 3.2 (37/1148) | |
| Age (years) [β] | 57, 57–64 | 58, 58–63 | 0.729[$] |
| <40 [δ] | 97.4 (267/274) | 2.6 (7/274) | 0.967 |
| 41–55 [δ] | 97.4 (530/544) | 2.6 (14/544) | |
| >55 [δ] | 97.2 (914/940) | 2.8 (26/940) | |
| Gender [δ] | | | 0.261 |
| Female | 97.7 (978/1001) | 2.3 (23/1001) | |
| Male | 96.8 (733/757) | 3.2 (24/575) | |
| Liver cirrhosis [δ] | | | 0.874 |
| Non cirrhosis | 97.3 (1105/1136) | 2.7 (31/1136) | |
| Compensated | 97.4 (597/613) | 2.6 (16/613) | |
| Decompensated | 100 (9/9) | | |
| Diabetes [δ] | | | 0.653 |
| No | 97.4 (1658/1703) | 2.6 (45/1703) | |
| Yes | 96.4 (53/55) | 3.6 (2/55) | |
| HBV coinfection [δ] | | | 0.797 |
| No | 97.3 (1664/1710) | 2.7 (46/1710) | |
| Yes | 97.9 (47/48) | 21. (1/48) | |
| HIV coinfection | | | 0.679 |
| No [δ] | 97.3 (1687/1733) | 2.7 (46/47) | |
| Yes [δ] | 96.0 (24/25) | 4.0 (1/25) | |
| BMI (kg/m2) [α] | 22.7 ±3.28 | 22.5 ± 2.86 | 0.752[$] |
| <18 [δ] | 100 (70/70) | 0 (0/0) | 0.276 |
| 18–25 [δ] | 96.7 (999/1033) | 3.3 (34/1033) | |
| >25 [δ] | 97.3 (291/299) | 2.7 (8/299) | |
| APRI [α] | 6.8 ± 23.8 | 6.9 ± 24.3 | 0.999[$] |
| ≥ 2 [δ] | 96.4 (80/83) | 3.6 (3/83) | 0.584 |
| < 2 [δ] | 97.4 (1524/1565) | 2.6 (41/1565) | |
| FIB-4 [α] | 8.9 ± 23.4 | 9.08 ± 23.9 | 0.704[$] |
| ≥ 3.5 [δ] | 97.7 (417/427) | 2.3 (10/427) | 0.625 |
| < 3.5 [δ] | 97.2 (1187/1221) | 2.8 (34/1221) | |
| Baseline HCV RNA (log IU/mL) [β] | 6.0, 6.0–6.6 | 6.3, 6.3–6.8 | 0.055[$] |
| ≤6000000 IU/mL [δ] | 98.1 (785/800) | 1.9 (15/800) | 0.058 |
| >6000000 IU/mL [δ] | 96.7 (926/958) | 3.3 (32/958) | |
| RVR achieved [δ] | | | 0.877 |
| Yes | 97.4 (1493/1533) | 2.6 (40/1533) | |
| No (≥15 IU/ml) | 97.6 (201/206) | 2.4 (5/206) | |
| Regimen [δ] | | | 0.164 |
| SOF/LDV± RBV | 97.1 (1437/1480) | 2.9 (43/1480) | |
| SOF/DAC± RBV | 98.6 (274/278) | 1.4 (4/278) | |
| Ribavirin [δ] | | | 0.312 |
| No | 97.0 (1116/1150) | 3.0 (34/1150) | |

*(Continued)*

**Table 2.** (Continued)

| Variable | SVR achieved n = 1711 % (n) | SVR failure n = 47 % (n) | P value |
|---|---|---|---|
| Yes | 97.9 (595/608) | 2.1 (13/608) | |
| Treatment time $^\delta$ | | | 0.071 |
| 12 weeks | 97.5 (1651/1694) | 2.5 (43/1694) | |
| 24 weeks | 93.8 (60/64) | 6.3 (4/64) | |
| Prior treatment failure $^\delta$ | | | 0.302 |
| No | 97.2 (1615/1661) | 2.8 (46/1661) | |
| Yes | 99.0 (96/97) | 1.0 (1/97) | |

$^\alpha$: mean ±SD;

$^\beta$: median; interquartile range;

$^\delta$: %(n)

$^\Omega$: Chi—square test;

$^\$$: Mann Whitney U test.

There was no significant difference in treatment failure rates between patients treated with SOF/LED ± RBV versus SOF/DAC± RBV (2.9%; 43/1480 versus 1.4%; 4/278 $p$ = 0.164 chi-squared test), or treated with or without RBV (n = 608) (2.1% (13/608) versus 3.0% (34/1150), $p$ = 0.312, chi-squared test). Treatment failure was more frequent among patients treated with a 24 week regimen compared to a 12 weeks regimen (6.3%; 4/64 versus 2.5%; 43/1694), although this difference did not quite reach statistical significance ($p$ = 0.071 chi-squared test). However, in patients infected with genotype 6, there appeared to be a marked increase in the rate of treatment failure in those receiving 24 weeks treatment rather than 12 (11.4% (4/35) versus 3.0% (33/1113); $p$ = 0.005, chi-squared test).

We further examined the nature of treatment failure; whether it is a breakthrough or relapse. In 1739 patients, where RVR results were available 88.2% (1533/1739) achieved a RVR. All patients with RVR failure had undetectable HCV RNA when measured at 8 week after IOT. Those who achieved RVR (n = 1533), 2.6% (40/1533) had either viral breakthrough 4 weeks after IOT or viral relapse after EOT. The rate of breakthrough or relapse was significantly higher in genotype 6 patients compared to genotype 1 (3.3%; 33/1005 versus 1.3%; 7/528, $p$ = 0.022 chi-squared test). Among 206 patients who failed to achieve a RVR, and subsequently had undetectable HCV RNA at 8 week IOT, only 2.4% (5/206) had treatment failure. The prevalence of treatment failure was not significantly different by virus genotype (1.3% genotype 1 (1 of 76) versus 3.1%; genotype 6 (4/130), $p$ = 0.428 chi-squared test) in these patients. Among patients without RVR data (n = 19), 89.5% (17/19) achieved a SVR and 10.5% (2/19) had treatment failure.

In patients with treatment failure (n = 47), the mean baseline viral load was higher than those who achieved SVR (6.7 ± 7.0 versus 6.4 ± 6.4 log IU/ml; $p$ = 0.055 Mann Whitney U Test). However, the majority of these patients attained RVR with viral relapse occurring 12 weeks after EOT.

A bivariate analysis was conducted to examine the effects of HCV genotype and baseline viral load on SVR or cure. Patients with genotype 1 had a higher probability of achieving cure (OR = 1.99; 95% CI: 0.98–4.04; $p$ = 0.054). Similarly, patients with lower viral loads had a higher probability of achieving SVR (OR = 1.8, 95% CI = 0.97–3.36, $p$ = 0.061) Table 3.

**Table 3. Odds ratio for achieving SVR in 1758 HCV patients treated with SOF/LED ± RBV or SOF/DAC ± RBV.**

| Variable | SVR rate | OR (95% CI) | P value |
|---|---|---|---|
|  | % (n) | Univariable |  |
| Age (years) |  |  | 0.967 |
| <40 | 97.4 (267/274) | 1.08 (0.46–2.52) | 0.850 |
| 41–55 | 97.4 (530/544) | 1.07 (0.55–2.08) | 0.825 |
| >55 | 97.2 (914/940) | 1.00 |  |
| Gender |  |  |  |
| Female | 97.7% (978/1001) | 1.39 (0.78–2.48) | 0.263 |
| Male | 96.8% (733/757) | 1.00 |  |
| Liver cirrhosis |  |  |  |
| Non cirrhosis | 97.3 (1105/1136 | 1.00 |  |
| Compensated | 97.4 (597/613) | 1.04 (0.56–1.92) | 0.884 |
| Decompensated | 100 (9/9) | NA |  |
| Diabetes |  |  |  |
| No | 97.4 (1658/1703) | 1.39 (0.32–5.88) | 0.654 |
| Yes | 96.4 (53/55) | 1.00 |  |
| HBV coinfection |  |  |  |
| No | 97.3 (1664/1710) | 1.00 |  |
| Yes | 97.9 (47/48) | 1.29 (0.17–9.62) | 0.798 |
| HIV coinfection |  |  |  |
| No | 97.3 (1687/1733) | 1.52 (0.20–11.53) | 0.681 |
| Yes | 96.0 (24/25) | 1.00 |  |
| BMI (kg/m2) |  |  |  |
| <18 | 100 (70/70) | NA |  |
| 18–25 | 96.7 (999/1033) | 0.80 (0.37–1.76) | 0.592 |
| >25 | 97.3 (291/299) | 1.00 |  |
| APRI | 6.8 ± 23.8 | 1.00 (0.98–1.01) | 0.978 |
| ≥ 2 | 96.4 (80/83) | 1.00 |  |
| < 2 | 97.4 (1524/1565) | 1.39 (0.42–4.59) | 0.586 |
| FIB-4 | 8.9 ± 23.4 | 1.00 (0.98–1.01) | 0.975 |
| ≥ 3.5 | 97.7 (417/427) | 1.19 (0.58–2.43) | 0.626 |
| < 3.5 | 97.2 (1187/1221) | 1.00 |  |
| Baseline HCV RNA (IU/mL) |  |  |  |
| ≤6000000 IU/mL | 98.1 (785/800) | 1.80 (0.97–3.36) | 0.061 |
| >6000000 IU/mL | 96.7 (926/958) | 1.00 |  |
| Genotype |  |  |  |
| Genotype 1 | 98.4% (600/610) | 1.99 (0.98–4.04) | 0.054 |
| Genotype 6 | 96.8 (1111/1148) | 1.00 |  |
| RVR achieved |  |  |  |
| Yes | 97.4 (1493/1533) | 1.00 |  |
| No (≥15 IU/ml) | 97.6 (201/206) | 1.07 (0.42–2.76) | 0.877 |
| Regimen |  |  |  |
| SOF/LDV± RBV | 97.1 (1437/1480) | 1.00 |  |
| SOF/DAC± RBV | 98.6 (274/278) | 2.05 (0.73–5.75) | 0.173 |
| Ribavirin |  |  |  |
| No | 97.0 (1116/1150) | 1.00 |  |
| Yes | 97.9 (595/608) | 1.39 (0.73–2.66) | 0.314 |
| Treatment time |  |  |  |

(*Continued*)

**Table 3.** (Continued)

| Variable | SVR rate | OR (95% CI) | P value |
|---|---|---|---|
| | % (n) | Univariable | |
| 12 weeks | 97.5 (1651/1694) | 2.56 (0.86–7.36) | 0.081 |
| 24 weeks | 93.8 (60/64) | 1.00 | |
| Prior treatment failure | | | |
| No | 97.2 (1615/1661) | 1.00 | |
| Yes | 99.0 (96/97) | 2.73 (0.37–20.04) | 0.322 |

## Discussion

We performed a retrospective review of HCV in our hospital in order to understand the response in patients infected with HCV genotype 6 to DAAs. Genotype 6 is the most frequent cause in our patients. There are a few data on the response of genotype 6 infections to DAAs, particularly in low-income settings. Our study addresses this knowledge gap and adds to the real-world data on the effectiveness of DAAs in genotype 6 in clinical practice [11, 12]. We compared treatment responses in patients infected with genotype 6 with those of patients infected with genotype 1 virus. We chose this comparison because i) current treatment guidelines recommend the same drug combinations can be used for each of these genotypes, and ii) the wealth of data from rich countries regarding the treatment response of genotype 1 infections allows us to set our experience in context. Our data add to the limited number of reports on treatment response that have emerged from Asian countries, including Vietnam [17, 23].

Similar to earlier studies, we documented a high prevalence (54.7% of cases) of HCV genotype 6 among patients attending our hospital [24]. This might be due to lower rate of spontaneous clearance of HCV genotype 6 than other genotypes or genotype 6 infections respond poorly to historical (non-DAA) anti-HCV therapy (e.g. PegINF±RBV) [18]. HCV genotype 6 is unique in many respects including i) localized geographic epidemiology (Laos, Cambodia, Vietnam, Myanmar, and Southern China), ii) high genetic diversity [9], iii) high number of preexisting drug resistance mutations [25], and iv) variable in-vitro susceptibility to DAAs (e.g. LDV) [25]. Compared with genotype 1 infections, we observed a higher prevalence of genotype 6 infection in women compared with men. The reasons for this is unclear, but it may represent inequalities in health care access between men and women in Vietnam. Historically treatment for HCV has been expensive and funded by the patient; fewer women may have had access to funds for treatment. The median age of patients in our study was 57 years. We found that patients with genotype 1 infections were younger then genotype 6 infected patients, and had higher rates of HBV and HIV coinfection. This points to some separation in the epidemics of genotype 6 and 1 infections in Vietnam. However, this being a retrospective study, we were unable to interrogate this further. It is feasible that the differences in genotype epidemics are associated with different risk behaviors (e.g. injectable drug use, man sex with man and sexual risk behavior) in the Vietnamese population at specific times. We found that patients with genotype 6 infections tended to have higher baseline viral loads than patients with genotype 1 infections. The consistency of this finding with previous studies suggests that genotype 6 virus may have a higher replication rate than genotype 1 virus [24].

Vietnamese guidelines at the time of this study suggested patients should receive either 12 weeks of treatment with or without RBV or 24 weeks of treatment without RBV depending upon the degree of their underlying liver disease. The majority of patients in our study received a 12 week treatment course. This is probably because the 12 week treatment course is

significantly cheaper than 24 weeks, although patient convenience and adherence may also have been part of clinical decision making [24]. The American Association for the Study of Liver Diseases (AASLD), the European Association for the Study of the Liver (EASL), and the Asian-Pacific Association for the Study of the Liver (APASL) recommend daily fixed-dose combination of i) glecaprevir/pibrentasvir for 8 weeks, ii) sofosbuvir/velpatasvir for 12 weeks, iii) SOF/LDV for 12 weeks, or iv) SOF/DAC for 12 weeks for treatment naïve HCV genotype 6 patients [26–28]. However, SOF/LDV is not currently recommended for treatment of subtype 6e infections[26]. There is a move towards recommending treatment durations for HCV infection (for example, as per AASLD guidelines); shorter durations of treatment can reduce costs and aid adherence. However, it is important that such recommendations for Asia are backed up by clinical trial evidence that includes patients with genotype 6 infections. Studies on treatment shortening are under evaluation in Vietnam.

We observed excellent cure rates ($>95\%$) in both patients with genotype 1 and genotype 6 HCV infections in our cohort, and cure rates were similar for both SOF/LDV and SOF/DAC combination therapy. We did not observed statistically significant difference in treatment outcomes between genotype 1 and 6 for these durations of treatment, although larger numbers of patients need to be evaluated to ensure this is true. We found cure rates of 96.8% (1111/1148) in genotype 6 infections versus 98.4% (600/610) in Genotype 1 ($p = 0.05$) infections. The slightly lower response we observed in genotype 6 infections could be explained by i) the higher baseline viral load in genotype 6 disease, ii) presence of pre-existing drug resistance mutations in genotype 6, and iii) the genetic diversity of genotype 6. A study from Myanmar found unexpectedly low SVR rates with sofosbuvir/ledipasvir combination treatment in genotype 6 infected patients [29]. In-vitro susceptibility studies have shown that there is variability amongst sub-lineages of genotype 6 virus to some DAAs. For example, *in vitro* resistance selection studies with LDV identified the single Y93H or Q30E resistance-associated variants (RAVs) in the NS5A gene in HCV genotype 6e. Similar RAVs were also observed in patients after a 3-day monotherapy treatment with LDV in genotype 1b [30]. Subtype 6e is also the predominant subtype in southern Vietnam. [18]. We do not have highly resolved genotype data for the infections in our study; however, the overall excellent response rates that we found suggest that either the treatment combinations used in our cohort are in fact highly effective across the genotype 6 subtypes, or subtype 6e is not frequent in our patients.

In our study, treatment failure was higher in patients treated with a 24-week regimen. This is possible because patients with cirrhosis (Child pugh B or C) are often treated with a 24-weeks regimen. These patients were older, had higher liver stiffness, high APRI and FIB-4 index and respond poorly to DAAs.

The vast majority of patients in our study had rapid virological responses, with undetectable viral loads by 4 weeks after treatment initiation. There was no difference in the rates of RVR by genotype. However, eventual cure rates were similar between patients who did and did not achieve RVRs, suggesting that viral load measured at this time point has little clinical utility where at least 12 weeks of treatment is prescribed and the patient is adherent. However, our experience contrasts with that of others where RVR has appeared to have a predictive value [31].

We could not determine whether the treatment failure was a result of relapse (SVR failure after EOT) or breakthrough (SVR failure during treatment) as viral load data at end of treatment was not available. However, based on the fact that 89.3% (42/47) treatment failure had RVR, one might speculate that most of the treatment failure were due to viral relapse after EOT or viral breakthrough 4 weeks after IOT. This suggests a possible adaptation/mutation in the viral genome or selection of resistant variants during the course of treatment. It has been reported that RVR and very rapid virologic response (vRVR; undetectable serum HCV RNA

level at week 2) has a high positive but low negative predictive value of SVR with dual sofosbu-vir/ribavirin therapy [32].

Our study is not without limitations. Our centre is a tertiary care centre and therefore the patients and outcomes may not be representative of the wider patient population in Vietnam. During the period of the study, HCV treatment was available only to self-funded patients. Given that DAA treatment was costing around $2500/patient at the time, it is likely that most patients are wealthy and therefore patients from lower socioeconomic groups may not be represented. It was not possible to interrogate this with the available dataset. Our liver status data may be biased as we could only analyse the data from patients who could afford the test. Our study includes only patients who have completed the treatment as patients with incomplete treatment or discontinued treatment lacked SVR viral load data.

In conclusion, genotype 6 infection appears to be the predominant infecting HCV genotype in the south of Vietnam. Treatment outcomes in our tertiary referral centre were largely comparable to those in rich developed countries when treated for 12 weeks. It is possible that G6 outcomes are slightly worse than genotype 1, but any differences are small. However, there remains a need to generate evidence from randomized control trials on the best treatment combinations and options for patients in Asia infected with genotype 6.

## Supporting information

**S1 Table. Treatment regimens and dose following Decision No. 5012/QĐ-BYT by MoH, Vietnam.** A; non cirrhotic chronic HCV, B; Chronic HCV with compensated cirrhosis (Child Pugh A), C: Chronic HCV decompensated cirrhosis (including moderate and severe liver failure, Child Pugh B or C), D: Doses of treatment.
(DOCX)

**S2 Table. Test recommendation before, during and after treatment of chronic HCV with DAA, DAA+RBV and PegINF+RBV+SOF (Issued together with Decision No. 5012/QĐ-BYT by MoH, Vietnam).**
(DOCX)

## Acknowledgments

We thank Oxford University Clinical Research Unit, Vietnam and Hospital for Tropical Diseases, Ho Chi Minh City, Vietnam for resources for Data analysis.

## Author Contributions

**Conceptualization:** Dung Thanh Nguyen, Motiur Rahman, Hung Mạnh Le.

**Data curation:** Thanh Thi Thanh Tran, Quang Minh Vo, Jeremy Day.

**Formal analysis:** Thanh Thi Thanh Tran, Ngoc My Nghiem, Quang Minh Vo.

**Investigation:** Dung Thanh Nguyen, Ngoc My Nghiem, Phuong Thanh Le.

**Methodology:** Dung Thanh Nguyen, Thanh Thi Thanh Tran, Ngoc My Nghiem, Phuong Thanh Le, Hung Mạnh Le.

**Project administration:** Ngoc My Nghiem, Phuong Thanh Le, Hung Mạnh Le.

**Resources:** Motiur Rahman.

**Software:** Thanh Thi Thanh Tran.

**Supervision:** Dung Thanh Nguyen, Phuong Thanh Le, Quang Minh Vo, Jeremy Day, Motiur Rahman, Hung Mạnh Le.

**Validation:** Jeremy Day.

**Writing – original draft:** Ngoc My Nghiem, Motiur Rahman, Hung Mạnh Le.

**Writing – review & editing:** Dung Thanh Nguyen, Phuong Thanh Le, Quang Minh Vo, Jeremy Day, Motiur Rahman, Hung Mạnh Le.

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
