## [Decision Letter · Decision Letter 0]

30 Mar 2020

PONE-D-20-06277

Effectiveness of sofosbuvir based direct-acting antiviral regimens for chronic hepatitis C virus genotype 6 patients: Real-world experience in Vietnam.

PLOS ONE

Dear Prof. Motiur Rahman,

Thank you for submitting your manuscript to PLOS ONE. After careful consideration, we feel that it has merit but does not fully meet PLOS ONE’s publication criteria as it currently stands. Therefore, we invite you to submit a revised version of the manuscript that addresses the points raised during the review process.

We would appreciate receiving your revised manuscript by May 14 2020 11:59PM. To enhance the reproducibility of your results, we recommend that if applicable you deposit your laboratory protocols in protocols.io, where a protocol can be assigned its own identifier (DOI) such that it can be cited independently in the future. For instructions see: http://journals.plos.org/plosone/s/submission-guidelines#loc-laboratory-protocols

We look forward to receiving your revised manuscript.

Kind regards,

Tatsuo Kanda, M.D., Ph.D.

Academic Editor

PLOS ONE

Journal Requirements:

2. In your ethics statement in the manuscript and in the online submission form, please provide additional information about the patient records used in your retrospective study. Specifically, please ensure that you have discussed whether all data were fully anonymized before you accessed them and/or whether the IRB or ethics committee waived the requirement for informed consent. If patients provided informed written consent to have data from their medical records used in research, please include this information.

Reviewers' comments:

Reviewer's Responses to Questions

**Comments to the Author**

1. Is the manuscript technically sound, and do the data support the conclusions?

Reviewer #1: Yes

Reviewer #2: Yes

2. Has the statistical analysis been performed appropriately and rigorously? 

Reviewer #1: Yes

Reviewer #2: Yes

3. Have the authors made all data underlying the findings in their manuscript fully available?

Reviewer #1: Yes

Reviewer #2: Yes

4. Is the manuscript presented in an intelligible fashion and written in standard English?

Reviewer #1: Yes

Reviewer #2: No

5. Review Comments to the Author

Reviewer #1: Authors conducted the real-world analysis aiming to evaluate the effectiveness of SOF based regimen in Vietnam. They demonstrated the overall SVR rates were 97.3% (1711/1758), and treatment failure ratios with different genotypes were: genotype 1, 1.7 % (10/610); genotype 6, 3.2% (37/1148). This study is well done and the manuscript is well written, but needs some modification for improvement.

1. How come the results among those patients with active and HCC post curative therapies?

Do they exclude "HCC"?

2. Author should refer and compare APASL/AASLD/EASL guideline and the treatment for GT-6 patients including other regimens.

3. The difference of RAVs between G-6e and other G-6 could be discussed. Also, the treatment and retreatment regimens for the G-6e patients could be discussed.

Reviewer #2: This original manuscript arouses interest for readers and provides an important clue to understand the epidemiology and properties of HCV genotype 6 and to treat patients with genotype 6 with DAAs. However, there are several issues that should be addressed or altered.

1) Where were results of the end-of-treatment response (ETR) rates? Authors should specify the ETR rates. If the HCV RNA negativity was unknown at the completion or premature cessation of treatment, one could not identify the treatment outcomes, viral breakthrough or relapse.

2) Have all subjects completed the 12-week or 24-week treatment regimens? How many patients ceased treatment prematurely?

3) Would you please inform us the patient and/or virological characteristics in more detail? Were there any differences in the characteristics between genotypes 1 and 6?

4) Line 153: the first “cirrhosis” is correct? Is the term “fibrosis”?

5) Lines 223 to 228: The descriptions in the text do not coincide with those in Table 1. Median or mean?

6) Lines 251 and 253: The descriptions do not coincide with those in Table 1.

7) Table 1: Which were two or three decimal places of decimals for p values? Table 1 should be more polished.

8) Line 264: “Rapid Virological Response” should be deleted from the text.

9) Line 279 should be re-written.

10) Table 2: Look at the cell in Liver cirrhosis, Decompensated, SVR failure.

11) Line 311: The parenthesis for “RVR” should be deleted from the text.

12) Lines 313, 314, and 315: The numbers are incorrect?

13) Line 401: The term is “difference”, but not “different”.

14) Lines 407 to 409: All the patients with treatment failure were relapsers. None had viral breakthrough. The ETR rates should be clearly described!

15) English language should be edited by a English-native speaker.

6. PLOS authors have the option to publish the peer review history of their article (what does this mean?). If published, this will include your full peer review and any attached files.

Reviewer #1: No

Reviewer #2: No

---

## [Author Response · Author response to Decision Letter 0]

3 May 2020

Response to editors comments:

Response: No changes are made on financial disclosure. 

To enhance the reproducibility of your results, we recommend that if applicable you deposit your laboratory protocols in protocols.io, where a protocol can be assigned its own identifier (DOI) such that it can be cited independently in the future. For instructions see: http://journals.plos.org/plosone/s/submission-guidelines#loc-laboratory-protocols

• A rebuttal letter that responds to each point raised by the academic editor and reviewer(s). This letter should be uploaded as separate file and labeled 'Response to Reviewers'.

Response to Reviewer’s is uploaded.

• A marked-up copy of your manuscript that highlights changes made to the original version. This file should be uploaded as separate file and labeled 'Revised Manuscript with Track Changes'.

Revised Manuscript with Track change is uploaded

• An unmarked version of your revised paper without tracked changes. This file should be uploaded as separate file and labeled 'Manuscript'.

A file labeled Manuscript is uploaded.

Journal Requirements:

Response: We have corrected the table and file name

2. In your ethics statement in the manuscript and in the online submission form, please provide additional information about the patient records used in your retrospective study. Specifically, please ensure that you have discussed whether all data were fully anonymized before you accessed them and/or whether the IRB or ethics committee waived the requirement for informed consent. If patients provided informed written consent to have data from their medical records used in research, please include this information.

Response: The study received ethical approval from the Ethics Review Committee of the Hospital for Tropical Diseases (approval no CS/ND/16/02 date 23/11/2017). The ethics committee have waived the requirement for informed consent and as recommended by IRB, all data were fully anonymized by third party before handing over to the analysis team. 

Response: The data presented in the manuscript are extracted from the Hospital for Tropical Diseases, Ho Chi Minh City Vietnam and property of Hospital for Tropical Diseases, Ho Chi Minh City Vietnam. Institutional Review Board (IRB) of the Hospital for Tropical Diseases approved the access to fully anonymized dataset for analysis to the investigators and Oxford University Clinical research Unit (OUCRU). OUCRU has established a data sharing policy and the data can be accessed through the data sharing policy. All data presented in the manuscript can be accessed through “OUCRU data sharing policy” and request for access to data can be sent to DAC@oucru.org. 

Reviewers' comments:

Reviewer's Responses to Questions

Comments to the Author

1. Is the manuscript technically sound, and do the data support the conclusions?

Reviewer #1: Yes

Reviewer #2: Yes

2. Has the statistical analysis been performed appropriately and rigorously? 

Reviewer #1: Yes

Reviewer #2: Yes

3. Have the authors made all data underlying the findings in their manuscript fully available?

Reviewer #1: Yes

Reviewer #2: Yes

4. Is the manuscript presented in an intelligible fashion and written in standard English?

Reviewer #1: Yes

Reviewer #2: No

5. Review Comments to the Author

Reviewer #1: Authors conducted the real-world analysis aiming to evaluate the effectiveness of SOF based regimen in Vietnam. They demonstrated the overall SVR rates were 97.3% (1711/1758), and treatment failure ratios with different genotypes were: genotype 1, 1.7 % (10/610); genotype 6, 3.2% (37/1148). This study is well done and the manuscript is well written, but needs some modification for improvement.

1. How come the results among those patients with active and HCC post curative therapies?

Do they exclude "HCC"?

Response: None of the patients included in the analysis had HCC

2. Author should refer and compare APASL/AASLD/EASL guideline and the treatment for GT-6 patients including other regimens.

Response: We have included reference and APASL/AASLD/EASL guideline for treatment of HCV genotype 6. Please see Page 23 line 372 – 378. We have also added necessary references (26-28).

3. The difference of RAVs between G-6e and other G-6 could be discussed. Also, the treatment and retreatment regimens for the G-6e patients could be discussed.

Response: We thank the reviewer for their comments. We have added the following text: For example, in vitro resistance selection studies with LDV identified the single Y93H or Q30E resistance-associated variants (RAVs) in the NS5A gene in HCV genotype 6e. Similar RAVs were also observed in patients after a 3-day monotherapy treatment with LDV in genotype 1b please see page 24, line 395 – 399.

Reviewer #2: This original manuscript arouses interest for readers and provides an important clue to understand the epidemiology and properties of HCV genotype 6 and to treat patients with genotype 6 with DAAs. However, there are several issues that should be addressed or altered.

1) Where were results of the end-of-treatment response (ETR) rates? Authors should specify the ETR rates. If the HCV RNA negativity was unknown at the completion or premature cessation of treatment, one could not identify the treatment outcomes, viral breakthrough or relapse.

Response: All patients are treated in accordance with the guidelines of the Vietnamese Ministry of Health for treatment of HCV infection. The guidelines mandate HCV viral load testing before initiation of treatment (IOT), at week 4 of treatment, again at week 8 if the patient had a detectable HCV viral load at week 4, and either 12 or 24 weeks after the end of treatment (EOT) (see S2 Table). The HCV viral load is not measured at the end of treatment and there therefore these data do not exist for any patient. Treatment cure is defined as an undetectable viral load measured between 12 and 24 weeks after completing treatment, and treatment failure as an HCV viral load ≥15 IU/ml 12 or more weeks after completing treatment). 

We have revised the statement on viral breakthrough or relapse in the results and discussion section. Please see page 18-19, line 305-316 and page 25 line 418-422. 

2) Have all subjects completed the 12-week or 24-week treatment regimens? How many patients ceased treatment prematurely?

Response: All patients completed 12-week or 24-week treatment. Patients with prematurely ceased treatment lacks SVR viral load data and was excluded from analysis. Please see page 6, line 116-117 and page 26 line 433-435. 

3) Would you please inform us the patient and/or virological characteristics in more detail? Were there any differences in the characteristics between genotypes 1 and 6?

Response: 

Tables 1 details differences between patients infected with HCV genotype 1 or 6. We found patients with genotype 6 infections were slightly older than those with genotype 1 infections (mean age) 55.87 years versus 53.20 years; p=<0.001, Mann Whitney U test). Overall there was a preponderance of female patients 56.9% (1001/1758). There was a preponderance of women amongst the genotype 6 infected cohort where they accounted for 59.8% of patients (95% confidence interval (CI) 57.0 - 62.6%; men 40.2%, 95%CI 37.4 – 43.0%, N = 1148). There was no difference in gender distribution amongst genotype 1 infections (women 51.5%, 95%CI 47.5 – 55.4%; men 48.5%, 95%CI 44.6 – 52.5%, N = 610). There was evidence of cirrhosis in 35.4% (622/1758) of patients and there was no difference in prevalence of liver cirrhosis between genotype 1 and 6 infected patients (p=.064, chi-squared test). There was a higher prevalence of HIV infection amongst patients with HCV genotype 1 infection than amongst patients with HCV genotype 6 infection (2.5% (15/610) versus 0.9% (10/1148) p=0.008, chi-squared test) patients. There was no signification difference in HBV coinfection among genotype 1 and 6 patients (2.8%; (17/610) versus 2.7% (31/1148) p=0.0531, chi-squared test). We found that markers of liver inflammation AST, ALT, AFP, GGT were statistically significantly higher in patients with genotype 1 infection, although the actual differences were small. In contrast, the HCV viral load was significantly higher in patients infected with genotype 6 virus compared with genotype 1 virus (6.6± 6.8 versus 6.3 ±6.5, p=<0.001, Mann Whitney U test)). There was no significant difference in APRI and FIB-4 scores between patients infected with genotype 1 versus genotype 6. Patients infected with genotype 1 were more likely to have had a prior treatment failure episode with PegINF/RBV (7.0% (43/610) versus 4.7% (54/1148); p=0.04, chi-squared test).We addressed this in the results page 11, line 222-242. 

4) Line 153: the first “cirrhosis” is correct? Is the term “fibrosis”?

Response: We have changed the term “cirrhosis” to “fibrosis”. Please see page 8 line 153.

5) Lines 223 to 228: The descriptions in the text do not coincide with those in Table 1. Median or mean?

Response: The data presented in Table 1 is median and interquartile range. In line 223 we have compared mean age. We have corrected the text by including “mean age”. Please see page 11, line 223.

6) Lines 251 and 253: The descriptions do not coincide with those in Table 1.

Response: We have corrected the description. Please see page 14, line 251 and 253.

7) Table 1: Which were two or three decimal places of decimals for p values? Table 1 should be more polished.

Response: we have presented all p values in three decimal. Please see table 1

8) Line 264: “Rapid Virological Response” should be deleted from the text.

Response: We have made the change. Please see page 14, line 264

9) Line 279 should be re-written.

Response: we have rephrased the sentence. Please see page 15, line 280 

10) Table 2: Look at the cell in Liver cirrhosis, Decompensated, SVR failure.

Response: we have made the correction, See Table 2 

11) Line 311: The parenthesis for “RVR” should be deleted from the text.

Response: We have made the change. Please see page 18, line 311-312

12) Lines 313, 314, and 315: The numbers are incorrect?

Response: We have corrected the numbers. Please see page 18, line 313-315 

13) Line 401: The term is “difference”, but not “different”. 

Response: We have made the change. Please see page 25, line 411

14) Lines 407 to 409: All the patients with treatment failure were relapsers. None had viral breakthrough. The ETR rates should be clearly described!

Response: We thank the reviewer for their comments. We have added the following text: We could not determine whether the treatment failure was a result of relapse (SVR failure after EOT) or breakthrough (SVR failure during treatment) as viral load data at end of treatment was not available. However, based on the fact that 89.3% (42/47) treatment failure had RVR, one might speculate that most of the treatment failure were due to viral relapse after EOT or viral breakthrough 4 weeks after IOT. Please see page 25, line 417-421.

15) English language should be edited by a English-native speaker.

Response: We have edited the English.

6. PLOS authors have the option to publish the peer review history of their article (what does this mean?). If published, this will include your full peer review and any attached files.

Do you want your identity to be public for this peer review? For information about this choice, including consent withdrawal, please see our Privacy Policy.

Reviewer #1: No

Reviewer #2: No

In compliance with data protection regulations, you may request that we remove your personal registration details at any time. (Remove my information/details). Please contact the p

---

## [Editor Report · Decision Letter 1]

6 May 2020

Effectiveness of sofosbuvir based direct-acting antiviral regimens for chronic hepatitis C virus genotype 6 patients: Real-world experience in Vietnam.

PONE-D-20-06277R1

Dear Dr. Motiur Rahman,

We are pleased to inform you that your manuscript has been judged scientifically suitable for publication and will be formally accepted for publication once it complies with all outstanding technical requirements.

With kind regards,

Tatsuo Kanda, M.D., Ph.D.

Academic Editor

PLOS ONE
---

## [Editor Report · Acceptance letter]

8 May 2020

PONE-D-20-06277R1 

Effectiveness of sofosbuvir based direct-acting antiviral regimens for chronic hepatitis C virus genotype 6 patients: Real-world experience in Vietnam. 

Dear Dr. Rahman:

I am pleased to inform you that your manuscript has been deemed suitable for publication in PLOS ONE. Congratulations! Your manuscript is now with our production department. 

With kind regards,

on behalf of

Dr. Tatsuo Kanda 

Academic Editor

PLOS ONE